# Tissue-Engineering the Fibrous Pancreatic Tumour Stroma Capsule in 3D Tumouroids to Demonstrate Paclitaxel Response

**DOI:** 10.3390/ijms22084289

**Published:** 2021-04-20

**Authors:** Judith Pape, Katerina Stamati, Rawiya Al Hosni, Ijeoma F. Uchegbu, Andreas G. Schatzlein, Marilena Loizidou, Mark Emberton, Umber Cheema

**Affiliations:** 1Centre for 3D Models of Health and Disease, Department of Targeted Intervention, Division of Surgery and Interventional Science, University College London, Charles Bell House, 43-45 Foley Street, London W1W 7TS, UK; judith.pape@ucl.ac.uk (J.P.); rawiya.hosni.14@ucl.ac.uk (R.A.H.); 2Research Department of Surgical Biotechnology, Division of Surgery and Interventional Sciences, Royal Free Hospital Campus, University College London, Rowland Hill Street, London NW3 2PF, UK; k.stamati@ucl.ac.uk (K.S.); m.loizidou@ucl.ac.uk (M.L.); 3Department of Pharmaceutical and Biological Chemistry, School of Pharmacy, University College London, 29-39 Brunswick Square, London WC1N 1AX, UK; ijeoma.uchegbu@ucl.ac.uk (I.F.U.); a.schatzlein@ucl.ac.uk (A.G.S.); 4Faculty of Medical Sciences, University College London, Maple House, 149 Tottenham Court Road, London W1T 7TNF, UK; m.emberton@ucl.ac.uk

**Keywords:** pancreatic cancer, desmoplasia, hyaluronic acid, chemoresistance, 3D tumour models

## Abstract

Pancreatic cancer is a unique cancer in that up to 90% of its tumour mass is composed of a hypovascular and fibrotic stroma. This makes it extremely difficult for chemotherapies to be delivered into the core of the cancer mass. We tissue-engineered a biomimetic 3D pancreatic cancer (“tumouroid”) model comprised of a central artificial cancer mass (ACM), containing MIA Paca-2 cells, surrounded by a fibrotic stromal compartment. This stromal compartment had a higher concentration of collagen type I, fibronectin, laminin, and hyaluronic acid (HA) than the ACM. The incorporation of HA was validated with alcian blue staining. Response to paclitaxel was determined in 2D MIA Paca-2 cell cultures, the ACMs alone, and in simple and complex tumouroids, in order to demonstrate drug sensitivity within pancreatic tumouroids of increasing complexity. The results showed that MIA Paca-2 cells grew into the complex stroma and invaded as cell clusters with a maximum distance of 363.7 µm by day 21. In terms of drug response, the IC_50_ for paclitaxel for MIA Paca-2 cells increased from 0.819 nM in 2D to 3.02 nM in ACMs and to 5.87 nM and 3.803 nM in simple and complex tumouroids respectively, indicating that drug penetration may be significantly reduced in the latter. The results demonstrate the need for biomimetic models during initial drug testing and evaluation.

## 1. Introduction

Pancreatic ductal adenocarcinoma (PDA) is one of the most devastating cancer prognoses due to the extremely poor five-year survival rates of 15–20% [1]. The disease’s development involves activating and somatic mutations in in the *K-ras*, *p16*, *p53*, and *DPC4* genes [2]. As PDA progresses, the staging is defined by tumour size and spread to lymph nodes, major local blood vessels, and other organs such as the spleen, lungs, or liver. Common treatments may involve surgery to remove parts of the pancreas (e.g., Whipple) or complete pancreatectomy [3]. Palliative radiotherapy may also be applied to shrink tumour mass. Chemotherapies such as gemcitabine, cisplatin, or paclitaxel are used for prolonged treatments [4]. Adjuvant chemotherapy may be used such as gemcitabine or 5-FU [5]. Whilst there are a range of treatments available for pancreatic cancer, only 4% of patients will live past five years after initial diagnosis [6]. A major reason for treatment failure is attributed to chemoresistance due to ongoing mutations caused by escape mechanisms within the cancer. Although genetic mutations drive tumour progression, there is a poor association with chemoresistance. There has been speculation that the epithelial-mesenchymal transition (EMT) [7] and CD44 positivity may contribute to this chemoresistance observed. The PDA stroma is very unique as it involves the trigger of an extensive desmoplastic reaction. It is also hypovascular, making drug delivery particularly difficult [8]. Secondly, the dense tumour stroma contributes to poor profusion of chemotherapies. The PDA tumour stroma, demonstrated in Figure 1, is a dense, fibrotic capsule forming around the abnormal cells enhancing stiffness and contractility [9]. This “capsule” is densely populated by pancreatic cancer-associated fibroblasts (CAFs) producing large amounts of fibronectin. Other extracellular matrix proteins present in high abundance are laminins and hyaluronic acid (HA). The collagen concentration of the tumour stroma is up to three times higher compared to the central tumour mass. The high abundance of HA, a non-sulfated glycosaminoglycan, contributes to bulk tumour mass. PDAs highly express CD44, a cell surface glycoprotein and receptor for HA. CD44 is largely involved in cell migration and angiogenesis explaining the accumulation of HA around the cancer [10]. HA also contributes to an increased level of interstitial fluid pressure (IFP), retaining water molecules with a high affinity [11].

Although 3D models of PDA such as spheroids have been used to model the cancer itself, it has become quite apparent that the tumour stroma may be just as, if not more, important in order to correctly model the disease. Recent advances in modelling PDA have highlighted the importance of modelling the tumour stroma. This can be seen in the novel use of 3D bioprinted platforms and microfluidic devices [12] in order to approach personalised drug testing [13]. It is increasingly accepted that the use of 3D cancer models is more relevant to study disease progression [14]. For PDA, spheroid models have been used to demonstrate the matrix-rich chemoresistance phenotype [15]. Some more complex 3D models utilising Matrigel® have also been described [16]. There has been limited work using collagen I and HA to create a biologically derived matrix as a 3D model [17].

The main aim of this work was to develop a biomimetic 3D model of pancreatic cancer and therefore create a pancreatic tumouroid, based on techniques we described previously [18]. Specifically, we created a two-compartment model consisting of a central artificial cancer mass (ACM), surrounded by a stiff tumour stroma containing laminin, fibronectin, HA, and increasing collagen densities. Tumouroids were utilised as a drug testing platform and calibrated against simple ACMs and cells grown in 2D. We interrogated the changes in drug response whilst the model increased in complexity representing chemical and physical barriers to drug penetration.

## 2. Results

### 2.1. Alcian Blue Staining for Presence of HA

In order to accurately model the fibrotic stroma in PDA, HA had to be incorporated into the stromal compartment of the tumouroids. Increasing concentrations were added to collagen gels prior to plastic compression. The indicative colorimetric data for the alcian blue stain concluded that the HA was successfully incorporated into the collagen matrix. As demonstrated in Figure 2, the alcian blue stained cartilage fully with cells visible whilst only staining the negative control (0 µg/mL HA) with minimal background and similarly within the 100 µg/mL HA samples. The 500 µg/mL HA was the only concentration that had a visible alcian blue stain, indicating the presence of HA.

### 2.2. Outgrowth into the 3D Pancreatic Tumour Stroma

Initial experiments showed that the MIA Paca-2 cells were successfully cultured within the model and invasive outgrowth occurred at the ACM-stromal edge. This was first evident within simple phase-contrast images taken at temporal time points as seen in Figure 3 below. It can be observed that compared to day 1, the cancer cells grow out as single cells and cell clusters, detaching from the original ACM. These invasive bodies or clusters were then stained for Phalloidin and DAPI (Figure 4A,B) and z-stacks were taken (Figure 4C,D). 

### 2.3. Proliferation and Distance of Invasion Measurements

Cell survival is crucial for future drug assays within the tumouroids. To establish MIA Paca-2 cell survival and measure proliferation within the simple compartmentalised tumouroids, a metabolic assay was performed. This would later serve as an outcome measure for the drug assay. Firstly, PrestoBlue^®^ was performed (Figure 5A), however, the readings showed that there was only a significant increase between day 7 and day 14 (*p* < 0.0001) with no differences observed between day 14 and day 21. This is due to the fact that PrestoBlue^®^ cannot completely penetrate the spheroids formed within the tumouroids. Alternatively, the CellTiter-Glo^®^ 3D assay was used. Initially, this assay was performed on tumouroids containing either a simple or complex stroma (Figure 5B). The data indicated that a complex stroma significantly decreased metabolic activity at days 7, 14, and 21 (*p* < 0.0001 for all). This was further validated with the distance in invasion measured (Figure 5C). The distance of invasion progressively increased over time when measured at days 7, 14, and 21 and averages for this can be taken from Table 1. The cells invaded a significantly greater distance into the simple stroma by day 21 (*p* = 0.0122).

### 2.4. Paclitaxel Drug Assay and Establishment of IC_50_ in 2D and 3D

Preliminary drug testing with paclitaxel was carried out in 2D in order to establish baseline IC_50_ values. Firstly, a dose response curve was developed for 500 or 1000 cells per well (Figure 6A). The IC_50_ for 500 cells/well was 0.189 nM and for 1000 cells/well was 2.64 nM (not shown). When conducting statistical analyses of multiple *t*-testing between the two dose response curves, the first discovery and significant difference (*p* = 0.000491) was established at the fourth lowest drug concentration of 0.3 nM of paclitaxel. Above this concentration, maximal cell death was observed in both cell densities. This relates to the fact that a higher drug concentration is needed for a higher cellular concentration but also underlines the fact that cell number and an increase in IC_50_ is not directly proportional.

Moving the assay into 3D, IC_50_ for MIA Paca-2 cells in ACMs was established. Since the seeding density would be higher in 3D at 10,000 cells/ACM, the drug concentrations used for the dose response curves were increased by 10,100 and 1000-fold. In order to validly use a dose response curve, the IC_50_ needs to have at least two drug concentrations above and below its occurrence and one concentration needs to be sufficient enough to kill the majority of cells. Simultaneously, the R^2^ should be as close to 0.9 as possible. As seen in Figure 6B, the IC_50_ and R^2^ values were 3.02 nM (R^2^ = 0.9738) for the 10-fold increase in drug concentrations. Going on these parameters, the 10-fold increase in drug concentration compared to the 2D assay was chosen for the complex tumouroid set-up going forward.

The final dose response curves were established from paclitaxel treatment on simple or complex tumouroids containing an ACM embedded into a stromal compartment. When the drug assay was performed on the simple tumouroids (Figure 6C), the IC_50_ was 5.87 nM (R^2^ = 0.9689). In comparison, within a complex stroma values similar to the ACMs at 3.803 nM (R^2^ = 0.9437) were obtained again. However, this time, 50,000 cells were used within the ACM as there would have been more space for outgrowth and therefore the values are not directly comparable. Nevertheless, the data underline how this complex tumouroid model of pancreatic cancer may be used for initial drug screening.

## 3. Discussion

This brief explorative study into the possibility of tissue-engineering a PDA model with a focus on the fibrotic stromal compartment led us to draw three strong conclusions. Firstly, we were able to account for the major extracellular matrix components present within the dense and fibrous pancreatic cancer stroma, establishing a novel pancreatic tumouroid. Secondly, we were able to validate invasion and growth within these pancreatic tumouroids. Lastly, the tumouroid model could be used to conduct paclitaxel drug testing, making it a candidate for future drug screening and patient-specific modelling.

The research and advances within the pancreatic cancer field have been limited over recent years with not many breakthroughs, especially in drug treatment with rising resistance. Whilst the majority of work is conducted in 2D, some pancreatic cancer models to study growth and invasion have been used in the form of organoids and 3D models [19]. Common cell lines used are PANC-1, BxPC-3, AsPC-1, Capan-1/3, PaTu II, and MIA Paca-2, the one used within this study [20]. Some of these models have used a collagen I and Matrigel® mix, presumably in order to account for some of the extracellular matrix proteins that would be needed [16]. 2D and 3D models have their advantages and disadvantages, a common feature being that the stroma has not been modelled in a compartmentalised model like this before [21]. Some 3D models have utilised co-cultures with CAFs in order to investigate HIF-1α pathways [22]. Again, these spheroids do not account for the stroma of the tumour. Preclinical PDA models are under development but are limited as they circumvent the stromal influences a pancreatic tumour would harbour [23].

In terms of drug testing, there are some drug screenings using spheroids of different pancreatic cancer cell lines [15]. Our model not only tests within a 3D set-up for the cells themselves but provides an extracellular matrix as well as the tumour stroma which has proven extremely important in drug resistance. CD44 positive cells are known to promote gemcitabine resistance [24]. As MIA Paca-2 cells express CD44 strongly, the use of these cells is valid in this scenario. An important discussion point is drug penetration within this model. Previous work demonstrated successful drug penetration within this dense collagen set-up. A study exploring doxorubicin treatment on osteosarcoma tumouroids demonstrated even drug penetration and dose response values [25]. Furthermore, the tumouroid set-up was recently used to test pazopanib action on renal cancer cells to cause cell death and endothelial network disruption [26]. This indicates that the tumouroid set-up is a valid model for drug testing; however, the novel fibrotic capsule engineered in this particular piece of work needs to be investigated further. The direct role of how the ECM components potentially block drug penetration and therefore shift IC_50_ values needs to be explored visually in future studies.

This particular piece of work has extensive future potential for continuing works. Firstly, the incorporation of a fibroblast population would be of values as they are known to not only densely populate the cancer stroma but additionally produce important fibrotic factors such as additional fibronectin. These fibroblasts could be either a cell line, or patient-specific CAFs to make the model personalised [27]. Different drugs could then be screened on the model in order to establish a range of IC_50_ values and determine what drugs would work best for this specific patient.

## 4. Materials and Methods

### 4.1. Cell Culture and Maintenance

The MIA Paca-2 cell line was obtained through ECACC (through Sigma-Aldrich, Dorset, UK) and cultured in DMEM at 4500 mg/L glucose and supplemented with 10% Foetal Calf Serum (FCS) (First Link, Birmingham, UK), 100 units/mL penicillin, and 100 µg/mL streptomycin (Gibco^TM^ through Fisher Scientific, Loughborough, UK). Cells were routinely cultured and passaged twice a week in a humidified incubator at 37 °C and 5% CO_2_ atmospheric pressure.

### 4.2. Monomeric Collagen Type I Extraction from Rat-Tail

Type I collagen in the monomeric form was extracted from rat-tails, provided by Dr. Anita Singhani through the Royal Veterinary College, London, UK. The rat-tails were kept at −80 °C until ready to be used. On average, 30 rat-tails were used per extraction. The rat-tails were sliced open with a scalpel to roll away the skin and remove the tendons with strong forceps and a tight rolling action towards the anterior part of the tail. The tendons were then placed in 0.5 M acetic acid (Sigma-Aldrich, Dorset, UK) at 4 °C for one week to dissolve. The resulting solution was then filtered through glass wool (Glaswarenfabrik Karl Hecht GmbH & Co. KG, Sondheim vor der Rhön, Germany) to remove remnants of the tendons and to be left with a gloopy collagen solution. Five litres of pH 7.2 12.5 mM sodium phosphate dibasic (16.7 g) and 11.5 mM sodium phosphate monobasic (9.0 g) (both Sigma-Aldrich, Dorset, UK) dialysis buffer was prepared in distilled H_2_O for each dialysis step. The collagen solution was filled into 10 K MWCO 22 mm SnakeSkin^TM^ dialysis tubing (Thermo Fisher Scientific, Loughborough, UK) and dialysed against the buffer at 4 °C whilst stirred for six days with a solution change every 48 h. The collagen should precipitate out of its solution and an opaque, white jelly-like mass should remain in the tubes. This solution was then centrifuged at 12,200 RPM for 10 min at 4 °C and then left to dissolve in 1.5 L of 0.15 M acetic acid overnight. The following day, 75.0 g of sodium chloride (Sigma-Aldrich, Dorset, UK) was added and the solution was left overnight on a stirrer at 4 °C. This solution was then centrifuged again with the same parameters and underwent dialysis again for one week. The contents of the tubing were then centrifuged again and the supernatant discarded. The pellet was covered with 70% EtOH for two days and centrifuged a final time, supernatant removed and transferred into a petri dish where the collagen was washed again with 70% EtOH. After the washes, the collagen was frozen to −80 °C and freeze-dried in order to lyophilise the collagen. The collagen was then dissolved in 0.6% acetic acid and to prepare collagen gels. Sterilisation, if necessary, can be done on this solution with 1:1 chloroform stirring under a fume hood for 48 h.

### 4.3. Fabrication of 3D Complex Tumour Models; “Tumouroids”

All ACMs were fabricated using monomeric Type I rat-tail collagen (First Link, Birmingham, UK) and the RAFT^TM^ protocol pages 8–9 (Lonza, Slough, UK) as previously described [28]. To summarise: 10X MEM (Sigma-Aldrich, Dorset, UK Sigma-Aldrich, Dorset, UK) was mixed with collagen and neutralising agent (N.A.) (17% 10 M NaOH (Sigma-Aldrich, Dorset, UK) in 1 M HEPES buffer Gibco^TM^ through Thermo Fisher Scientific, Loughborough, UK)) and mixed with cell suspension resulting in 80% collagen, 10% 10× MEM, 6% N.A., and 4% cells. To seed the ACMs, 1 × 10^4^ cells for initial drug screening or 5 × 10^4^ cells for later tumouroids were added to 240 µL of the collagen mix per ACM and set in a 96-well plate (Corning^®^ Costar^®^ through Sigma-Aldrich, Dorset, UK). The gel mix was polymerised at 37 °C for 15 min, followed by plastic-compression using the 96-well RAFT^TM^ absorbers (Lonza, Slough, UK). In order to produce simple or complex “tumouroids” [18], the ACMs were nested into a stroma. For the simple stroma, collagen solution as described above was prepared, and ACMs were directly embedded into a 24-well plate (Corning^®^ Costar^®^ through Sigma-Aldrich, Dorset, UK) containing 1.3 mL of the non-cross-linked collagen mix. The tumouroids were again polymerised at 37 °C for 15 min and plastic-compressed using the 24-well RAFT^TM^ absorbers (Lonza, Slough, UK). Tumouroids were continuously cultured at 5% CO_2_ atmospheric pressure and 37 °C with 50% media changes every 48 h.

### 4.4. Incorporation of Laminin, Fibronectin, HA and Increasing the Collagen Concentration to Model the Complex Pancreatic Stroma

Extracellular matrix components were incorporated into the stromal compartment to establish complex tumouroids. These were in the form of: laminin, fibronectin, and HA. Mouse laminin at 50 µg/mL [29] (Corning^®^ through Sigma-Aldrich, Dorset, UK), human fibronectin at 25 µg/mL [30] (Sigma-Aldrich, Dorset, UK), and >1.8 MDa sodium hyaluronate (NaHy) at 500 µg/mL (Lifecore^TM^ Biomedical, Chaska, MN, USA) were added to the collagen mix that made up the stromal compartment. The collagen concentration was also increased by using a 6 mg/mL solution made up with the rat-tail collagen extracted as previously described.

### 4.5. Immunofluorescent Staining and Imaging

Tumouroids were formalin fixed using 10% neutrally buffered formalin (Genta Medical, York, UK) for a minimum of 30 min and then washed and stored in phosphate buffered saline (PBS) (Gibco^TM^ through Thermo Fisher Scientific, Loughborough, UK). The tumouroids were permeabilised and blocked overnight at room temperature using a solution of 0.2% Triton ×100 and 1% bovine serum albumin (BSA) (both Sigma-Aldrich, Dorset, UK) in PBS. In order to stain for Phalloidin, the Alexa Fluor^TM^ 568 Phalloidin kit was used in combination with DAPI, using NucBlue^TM^ (both Invitrogen^TM^ through Thermo Fisher Scientific, Loughborough, UK).

### 4.6. Measurement of Distance of Invasion

All tumouroid samples were imaged on the Zeiss AxioObserver with ApoTome.2 and Zeiss ZEN software (Zeiss, Oberkochen, Germany). In order to measure the invasion from the original ACM into the stromal compartment, four images were taken as previously described [28]. All samples were assessed for distance of invasion into the stromal compartment. The images obtained were analysed in Fiji ImageJ software [30].

### 4.7. Processing of Histological Samples and Alcian Blue Staining

The samples were formalin fixed using 10% neutrally buffered formalin (Genta Medical, York, UK) and processed by the histopathology department at the Royal National Orthopaedic Hospital, Stanmore, UK. The samples were wax-embedded and 5 µm sections were cut. Alcian blue staining was conducted on sectioned samples in order to visualise that the Sodium Hyaluronate (NAHy) was incorporated into the collagen and not removed during the process of plastic compression. After sectioning, a 500 µg/mL, 100 µg/mL, and 0 µg/mL HA gel was deparaffinized and hydrated to distilled water and stained with Alcian Blue (pH 2.5) (all Sigma-Aldrich, Dorset, UK) together with a positive control of porcine cartilage tissue. The samples were then washed extensively in distilled H_2_O, dehydrated, and imaged for colorimetric analysis.

### 4.8. Establishment of IC_50_ for Paclitaxel

For the drug assays and establishment of IC_50_ in response to the chemotherapy drug paclitaxel, a drug assay set-up for MIA Paca-2 was used first in 2D and then moved to 3D. For the 2D set-up, 500 or 1000 cells were seeded in 96-well plates and treated with seven increasing concentrations of paclitaxel dissolved in media containing 0.2% DMSO; 0.03 nM, 0.1 nM, 0.3 nM, 1 nM, 3 nM, 10 nM, and 30 nM (all Sigma-Aldrich, Dorset, UK). Cells were seeded and left to attach for 24 h followed by two doses at 72 h intervals. Metabolic activity was measured with PrestoBlue^®^ (Thermo Fisher Scientific, Loughborough, UK). For the 3D drug assay, ACMs containing 1 × 10^4^ cells were initially used to test the drug effect in 3D by increasing the drug concentrations by 10,100 and 1000-fold as compared to the 2D drug assay. The metabolic activity was then measured with CellTiter-Glo^®^ 3D Viability Assay (Promega, Southampton, UK). For the complex tumouroid set-up, 5 × 10^4^ cells were seeded per ACM. For both 3D assays, samples were left for 7 days to mature and then dosed twice with 72 h intervals.

### 4.9. Statistical Analyses

All statistical analyses were performed in GraphPad Prism 7/8 software. Data were tested for normality with the Shapiro–Wilk test (*n* ≥ 3) or the D’Agostino test (*n* ≥ 8) and the appropriate test for statistical significance was applied. The tests used for each data set are outlined within the figure legends. Significance was regarded at *p*-values < 0.05. All data points are represented as mean ± standard error mean (SEM) in graphs, tables, and text. In general, *n* = 3 with three technical replicates; details are described within the figure legends.

## Figures and Tables

**Figure 1 ijms-22-04289-f001:**
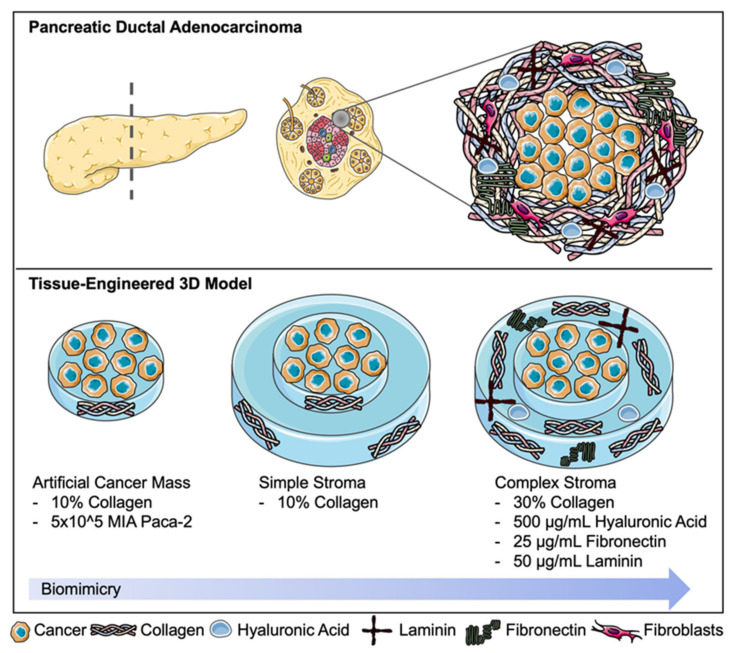
Pancreatic ductal adenocarcinoma and its desmoplastic, hypovascular tumour stroma. This fibrotic capsule contains a high concentration of extracellular matrix (ECM) components such as collagen, hyaluronic acid, laminins, and fibronectin partially produced by a dense population of cancer-associated fibroblasts. This schematic focuses on ECM components within the 3D tumouroid model. Created using Servier Medical Art according to the Creative Commons Attribution 3.0 Unported License guidelines 3.0 (https://creativecommons.org/licenses/by/3.0/, accessed on 9 February 2021).

**Figure 2 ijms-22-04289-f002:**
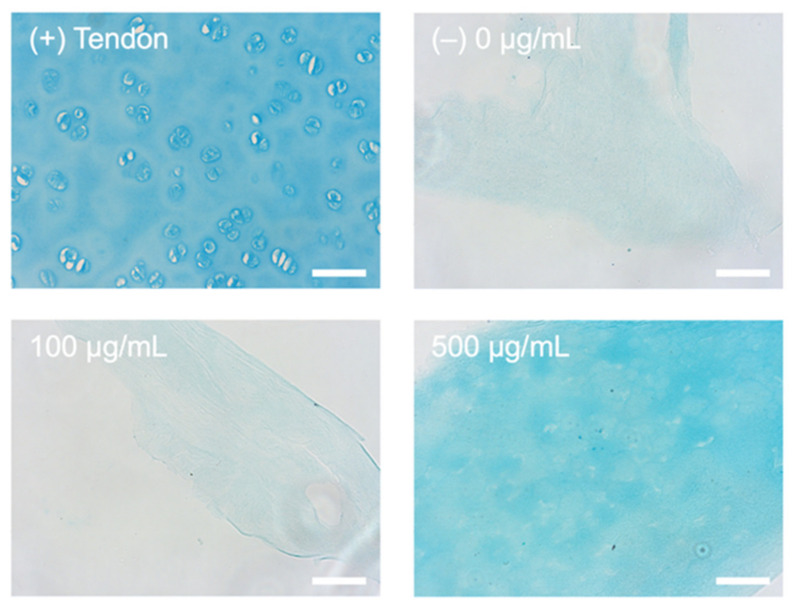
Alcian blue staining (blue) HA incorporation. Strong alcian blue staining is seen within the (+) cartilage control and the collagen gel with a concentration of 500 µg/mL HA. Light/background staining is seen in the HA 100 µg/mL HA and negative (−) 0 µg/mL HA control. Scale bar = 50 µm.

**Figure 3 ijms-22-04289-f003:**
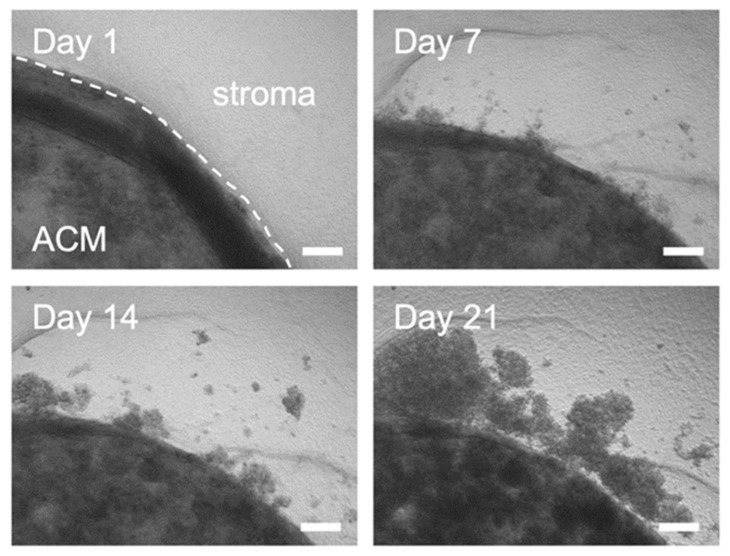
Phase contrast images of outgrowth into the stromal compartment over time. Images for longitudinal time points of MIA Paca-2 cell outgrowth from the artificial cancer mass (ACM) into the simple stroma. Scale bar = 100 µm.

**Figure 4 ijms-22-04289-f004:**
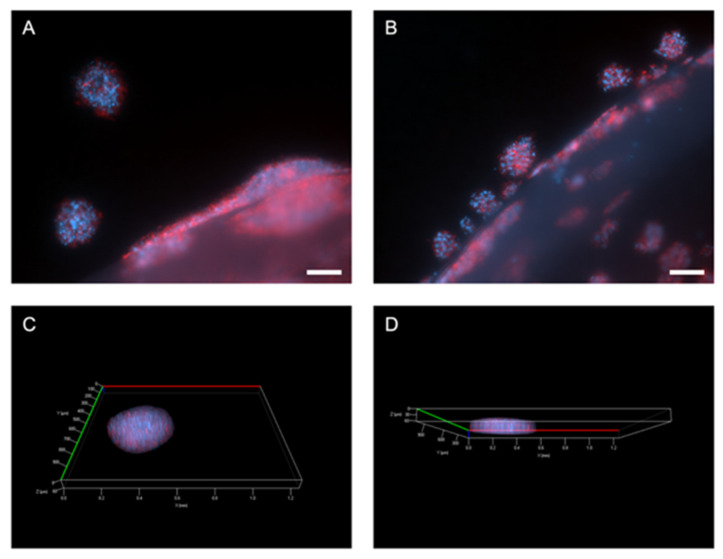
Outgrowth from ACM into stroma of pancreatic tumouroids. (**A**,**B**) Invasive outgrowth and invasive bodies growing out from the ACM into the stromal compartment. Scale bar = 100 µm. (**C**,**D**) Z-stacks of invasive bodies within the acellular stroma showing their volume. Red = Phalloidin and blue = DAPI.

**Figure 5 ijms-22-04289-f005:**
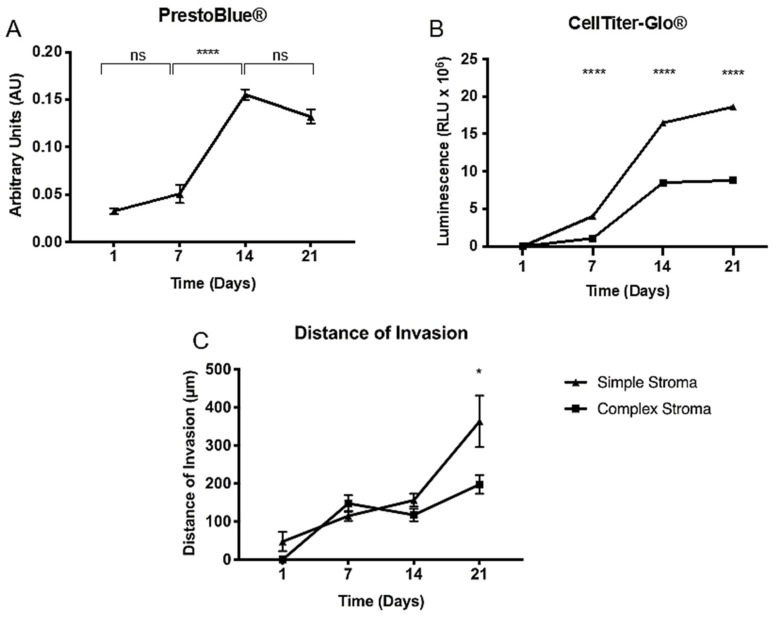
Metabolic activity measured in pancreatic tumouroids. (**A**) PrestoBlue^®^ and (**B**) CellTiter-Glo^®^ readings for MIA Paca-2 tumouroids over days 1, 7, 14, and 21. Values are (mean ± SEM) with *n* = 3 and three technical repeats and unpaired *t*-test with *p*-values ≤ 0.00005 **** for (**A**,**B**). (**C**) Distance of invasion from the MIA Paca-2 ACM into the fibrous stromal compartment. Values are (mean ± SEM) with *n* = 3 and three technical repeats. Mann–Whitney *p*-value ≤ 0.05 *.

**Figure 6 ijms-22-04289-f006:**
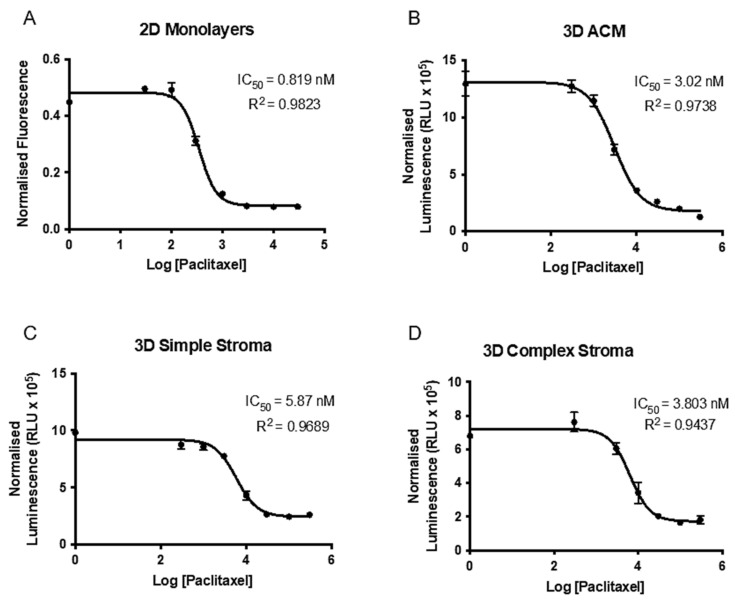
Dose response curve to increasing paclitaxel concentrations in 2D and 3D.(**A**) Seven increasing paclitaxel concentrations were used in 500 cells/well in a 2D set-up to establish initial IC_50_ values for MIA Paca-2 cells. (**B**) Dose response curve to increasing paclitaxel concentrations in 3D MIA Paca-2 ACMs and tumouroids with a (**C**) simple stroma or (**D**) complex stroma. Data represent (mean ± SEM) with *n* = 3 and three technical repeats. Values were normalized to DMSO vehicle control, transformed, and a non-linear fit was applied.

**Table 1 ijms-22-04289-t001:** Distance of invasion (µm) mean ± SEM from the ACM into the simple and complex stromal compartment.

	Day 1	Day 7	Day 14	Day 21
**Simple**	47.28 ± 25.42	114.8 ± 13.27	156.2 ± 16.95	279.0 ± 67.66
**Complex**	0.00 ± 0.00	147.8 ± 21.76	117.4 ± 16.18	197.6 ± 25.54

## Data Availability

All data sets are available upon reasonable request from the corresponding author.

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
