# Peer review of "Tissue-Engineering the Fibrous Pancreatic Tumour Stroma Capsule in 3D Tumouroids to Demonstrate Paclitaxel Response"

_ijms, 2021, doi:10.3390/ijms22084289_

Round 1

Reviewer 1 Report

In this manuscript, the authors present the fabrication of a biomimetic 3D model of pancreatic cancer containing MIA Paca-2 cells surrounded by a complex stromal compartment similar to the one found in vivo consisting of laminin, fibronectin, HA, and collagen. The study has great scientific importance as it deals with one of the most aggressive types of cancer. The tumouroid model presented here is an innovative platform for testing drug responses but also could be explored as a tool for drug development that could lead to more significant and relevant achievements in pancreatic cancer research.

The results demonstrated the successful incorporation of HA (500 μg/mL) within collagen, the invasiveness of tumor cells in the stromal compartment, the metabolic activity of cells during the time, and the dose-response to paclitaxel in 2D and 3D models.

The manuscript is well-organized and written, the methods are properly chosen and the conclusions of the research work are clearly formulated. I have no comments. The paper is suitable for publication in IJMS.

Author Response

We thank the reviewer for their positive feedback on the manuscript. We are happy they agree for the paper to be published. 

Reviewer 2 Report

In this manuscript the Authors report the composition of a 3D cancer model and its application on the efficacy evaluation of a known drug.

The manuscript is interesting and address a topic of special interest for (pre)clinical cancer research. Indeed, novel advanced models are fundamental to both push forward the translation of innovative treatments to patients and the biological knowledge of neoplasms. Overall, the manuscript is enough significant and novel to deserves publication in this Journal after the following minors are addressed:

  • To increase the impact of this manuscript, the Authors should add some biological assays (such as PCR, TEM etc.) to investigate if the 3D conformation of the model induces modifications in the cell behaviors.
  • The Authors should better discuss the introduction of the manuscript by considering the recent advances of the field (for example, doi: 10.1080/17425255.2021.1879047 and related).

Author Response

We thank the reviewer for their positive feedback and would like to address the two minor points made. 

1. We agree that it is very important to interrogate how the 3D conformation changes cell behavior. We have done extensive research into this question within the group over the years in colorectal, ovarian, osteosarcoma, and breast cancer. We have found that cells become more aggressive when they are cultured in 3D and lose expression of cytokeratins in some cases. We have also found that drug response improves in 3D compared to 2D. If time allowed, we would have looked at some gene markers for this also in the pancreatic cancer model. This is something that definitely should be investigated extensively in the future and we hope the reviewer can accept this as future works in this case. This paper was intended to be a short communication on how our model can be used as a drug testing platform. We hope to in the future investigate the pathways of pancreatic cancer in 3D at a larger scope.

2. We agree with the reviewer that the recent advances in 3D modeling should be outlined within the introduction. We have therefore included two more citations at the end of the introduction summarising the recent advances within the field.